# Correlation of Salivary Occult Blood with the Plasma Concentration of Branched-Chain Amino Acids: A Cross-Sectional Study

**DOI:** 10.3390/ijerph19158930

**Published:** 2022-07-22

**Authors:** Maya Izumi, Kazuo Sonoki, Sumio Akifusa

**Affiliations:** School of Oral Health Sciences, Faculty of Dentistry, Kyushu Dental University, Kitakyushu 803-8580, Japan; r15izumi@fa.kyu-dent.ac.jp (M.I.); sonoki@kyu-dent.ac.jp (K.S.)

**Keywords:** branched-chain amino acids, periodontitis, salivary occult-blood test, type 2 diabetes

## Abstract

Background: Plasma branched-chain amino acids (BCAA) levels are predictors of glycometabolic disorders, leading to diabetes. Microbes, including periodontal pathogens, are thought to be associated with elevated plasma BCAA levels. This study aimed to evaluate the relationship between salivary occult blood (SOB) and plasma BCAA levels in middle-aged Japanese individuals. Methods: Sixty-four Japanese individuals aged ≥ 40 years were recruited for this study, which was conducted in Fukuoka Prefecture, Japan, from August to December 2021. Individuals diagnosed with and/or treated for diabetes were excluded from the study. The body mass index (BMI); plasma concentrations of total, high-density, and low-density lipoprotein cholesterol; triglyceride, glucose, and BCAA; and glycosylated hemoglobin ratio were measured. A basic periodontal examination was performed after the SOB test. Results: The median age of participants (men—20; women—44) was 55 (range, 41–78) years. The plasma BCAA concentration in the SOB-positive group (477 [400–658] μmol/L) was higher than that in the SOB-negative group (432 [307–665] μmol/L). Linear regression analysis revealed that SOB remained independently associated with the plasma BCAA level with statistical significance (β = 0.17, *p* = 0.02) after adjusting for sex, age, and BMI. Conclusions: SOB was positively correlated with plasma BCAA levels in middle-aged Japanese individuals. Thus, SOB may be a predictor of elevated plasma BCAA levels.

## 1. Introduction

Previous studies strongly suggest a bidirectional relationship between type 2 diabetes and periodontitis [1,2,3,4]. A recent meta-analysis demonstrated an odds ratio (OR) of 1.85 for diabetes in patients with periodontitis [5]. Another meta-analysis revealed that the prevalence of type 2 diabetes was higher in patients with periodontitis (OR = 4.04, *p* < 0.001), and conversely, periodontitis prevalence was higher in patients with type 2 diabetes (OR = 1.58, *p* < 0.001) [6]. Further, inflammatory mediators associated with periodontitis increase the risk of insulin resistance, causing dysglycemia [7,8], while periodontal therapy can improve insulin resistance by reducing circulating inflammatory mediators, such as C-reactive protein, tumor necrosis factor, interleukin-6, and fibrinogen, in patients with type 2 diabetes [9]. Recent epidemiological and experimental studies have demonstrated that plasma branched-chain amino acids (BCAA; leucine, isoleucine, and valine) account for 40% of all plasma amino acids and are strongly associated with insulin resistance in humans and rodent models [10,11,12,13,14]. BCAA-catabolism defects lead to the elevation of BCAA levels and branched-chain α-keto acids, which are closely related to obesity-induced insulin resistance [15]. It is suggested that BCAA exacerbates insulin resistance either through the mammalian target of rapamycin-mediated suppression of insulin receptor substrate-1, or through the accumulation of toxic BCAA catabolites [7]. Longitudinal studies have demonstrated that high plasma BCAA levels are predictive of the future onset of type 2 diabetes [15,16,17]. BCAAs are essential amino acids, and the majority of BCAAs are derived from protein-containing foods, such as tuna, bonito, or chicken. However, recent studies have revealed that the intestinal flora is involved in the chronic elevation of plasma BCAA levels, and that two species of intestinal bacteria, *Prevotella copri*, and *Bacteroides vulgatus*, are associated with the induction of insulin resistance by elevating plasma BCAA levels in humans and rodent models [18,19]. A recent study demonstrated that the periodontal pathogen *Porphyromonas gingivalis* induces insulin resistance by increasing plasma BCAA levels in a rodent model [20]. While wild-type *P. gingivalis* enhances high-lipid-diet-induced insulin resistance, the BCAA aminotransferase-deficient *P. gingivalis* strain was unable to increase the plasma BCAA level and induce insulin resistance. The evidence presented above suggests that oral microbes may increase plasma BCAA levels. Considering that BCAA levels are a predictor of insulin resistance and glycemic dysbolism, we hypothesized that the prevalence of periodontal disease might be correlated with plasma BCAA levels in humans. In a previous study, poor periodontal conditions were associated with salivary occult blood (SOB) [21]. In this study, we tried to evaluate the periodontal status using a paper strip coated with anti-human-hemoglobin monoclonal antibody to detect SOB [21]. Therefore, we evaluated the correlation between indicators of periodontitis and plasma BCAA levels in middle-aged individuals to investigate the relationship between periodontal status and the risk of type 2 diabetes.

## 2. Materials and Methods

### 2.1. Study Setting and Study Population

This cross-sectional study conducted from August to December 2021 included 64 participants, who were the staff of 11 nursing-care insurance facilities in Fukuoka Prefecture, Japan. The inclusion criterion for the study was age ≥ 40 years. Individuals diagnosed with and/or treated for diabetes were excluded from the study. This study was approved by the human subjects ethics board of Kyushu Dental University Institutional Review Board for Clinical Research (No. 20-63) and was conducted in accordance with the Helsinki Declaration of 1975, as revised in 2013. Informed consent was obtained from all participants before data collection commenced. Strengthening the Reporting of Observational Studies in Epidemiology (STROBE) guidelines were followed for data analysis in this study (Appendix A).

### 2.2. Data Collection

Body height and weight were measured without shoes, and BMI (kg/m^2^) was calculated as an indicator of obesity. Venous blood samples were drawn at the cubital fossa after an overnight fast of at least 12 h. All blood samples were transported to the same laboratory in Fukuoka Prefecture (Hoken Kagaku, Inc., Kanagawa, Japan) for analysis. Plasma concentrations of total cholesterol, high-density lipoprotein (HDL) cholesterol, low-density lipoprotein (LDL) cholesterol, triglycerides, glucose, BCAA, and glycosylated hemoglobin (HbA1c) were measured using standard laboratory procedures. All clinical oral health examinations (number of present teeth, bleeding on probing [BOP], and periodontal pocket depths [PPDs]) were performed by a trained dentist (S.A.) from 1:30 p.m. to 3:30 p.m. at each facility. PPDs were assessed at the mesial, mid, and distal points of the buccal and lingual sides of each tooth. To distinguish periodontal disease, the number and ratio (%) of teeth with BOP and the number of teeth with PPD ≥ 4 mm were assessed [21]. Smoking habit was also collected.

### 2.3. SOB Test

The SOB test was performed on all participants in the sitting position, using Perioscreen^®^ (Sunstar Inc., Osaka, Japan), before the oral health examination [21]. Samples were collected by rinsing the participants’ mouths with 3 mL of water for 10 s and then spitting in a small paper cup. The lower end of the test strip was dipped in the sample. After 5 min, an examiner (S.A.) visually ranked the magenta-stained band according to a color chart as follows: no visible band = negative; a visible magenta band = positive. The manufacturer's reference for Perioscreen^®^ states that the magenta-stained band indicates the presence of human hemoglobin ≥ 2 μg/mL.

### 2.4. Sample Size

The sample size was not estimated at the start of the study.

### 2.5. Statistical Analysis

Descriptive statistics were used to characterize the participants. Since the data were not normally distributed, all values were represented as median (minimum–maximum), and the significance of the differences between the groups was evaluated using the Mann–Whitney U test for continuous variables and the chi-square test for categorical variables. Correlations between the glucose, HbA1c, total cholesterol, HDL-cholesterol, LDL-cholesterol, triglyceride levels, and BMI were tested using Spearman’s rank correlation test. Linear regression analysis was used to estimate the crude and adjusted models with the respective partial regression and standardized coefficients between SOB and BCAA. Sex, age, and BMI (to consider the effect of obesity) were added to the adjusted model. All analyses, excluding sample size, were carried out using SPSS statistical software version 28 (SPSS, Chicago, IL, USA), and the level of significance was set at 5% for all analyses.

## 3. Results

Data were obtained from 64 participants (20 males and 44 females). The median age of the participants was 55 (41–78) years; none of the participants were current smokers. Table 1 shows the characteristics of the participants in the SOB-negative and SOB-positive groups. In the SOB-positive group, the number and ratio (%) of teeth with BOP and PPD ≥ 4 mm were significantly higher than those in the SOB-negative group. No significant differences were observed in terms of sex and age. Diabetes-related variables, including plasma concentrations of total, HDL, and LDL cholesterol; triglycerides; fasting glucose; and HbA1c, were not significantly different between the groups. However, the plasma concentration of BCAAs in the SOB-positive group (477 [400–658] μmol/L) was significantly higher than that in the SOB-negative group (432 [307–665] μmol/L).

Although plasma BCAA concentrations were correlated with HbA1c (r = 0.30, *p* < 0.001), HDL cholesterol (r = −0.35, *p* = 0.006), and triglyceride (r = 0.44. *p* = 0.001) levels, they were not correlated with fasting blood glucose, BMI, or LDL cholesterol levels (Table 2).

Furthermore, we investigated whether SOB was a predictor of increased BCAA levels using linear regression analysis. Table 3 shows that even in the adjusted model, SOB remained independently and positively associated with plasma BCAA levels with statistical significance (β = 0.17, *p* = 0.020). The sample size for linear regression analysis was calculated using G* Power version 3 [22]. Based on an effect size f2 = 0.2, an α error = 0.05, 1 − β error = 0.8, and the number of predictors = 3, the required sample size for linear regression analysis was calculated as 59, which confirmed that the sample size was adequate for the analysis.

## 4. Discussion

In this cross-sectional study of middle-aged Japanese individuals, SOB was a possible predictor of increased plasma BCAA levels. To the best of our knowledge, this is the first report of the correlation between SOB and plasma BCAA levels in humans. Because *P. gingivalis* is regarded as a source of BCAA [20], our findings suggest that periodontal pathogen-derived BCAA may cause elevated plasma BCAA levels via periodontal lesions. Interestingly, *Streptococcus mutans* possesses BCAA aminotransferase activity encoded by *iLvE* [23]. *iLvE* plays a role in the biosynthesis and catabolism of BCAA [24,25] and in *S. mutans*’s acid tolerance [23] and survival after immune-cell phagocytosis [26]. According to the evidence presented above, oral bacteria that synthesize BCAA are not only associated with periodontal pathogenesis, but may also be involved in the elevation of plasma BCAA levels. Further epidemiological studies on the role of the oral microflora in the upregulation of BCAA are required in the future.

In this study, SOB was closely correlated with periodontal parameters, including the number of teeth with BOP, PPD ≥ 4 mm, and % BOP, which supports the findings in our previous study that the sensitivity and specificity of the SOB test for screening poor periodontal status, defined as the proportion of teeth with BOP ≥ 15% or the presence of teeth with PPD ≥ 4 mm, was 0.72 and 0.52, respectively [21]. Moreover, another study demonstrated that the sensitivity and specificity for >30% BOP was 75.9% and 90.5%, respectively [27]. These findings suggest that the SOB test is an easy, low-cost, rapid, and highly reliable screening method for initial-stage periodontal disease.

In this study, the plasma BCAA level was correlated with the ratio of HbA1c, but not fasting blood glucose. In this study, only the fasting plasma glucose level was assessed, and the postprandial and random glucose levels were not assessed. A recent study including a young healthy Chinese population demonstrated that the fasting and postprandial plasma BCAA concentrations were correlated with the postprandial glucose level, but not with the fasting glucose level [28]. To investigate the relationship between plasma BCAA and glucose levels, a postprandial sample may be needed. In obese and diabetic individuals, BCAA-catabolism defects contribute to the development of insulin resistance and diabetes [29,30,31]. An increase in intracellular glucose suppresses the expression of BCAA-degradation enzymes, resulting in BCAA accumulation [29]. A recent study revealed that the effect of BCAA on glucose metabolism is different between lean and obese mice [32]. In lean mice, BCAA-catabolism defects—resulting in BCAA accumulation in cells—lead to lower body weight and better glucose tolerance. The evidence shown above should be considered while generalizing our present findings.

In this study, the plasma BCAA concentration was positively correlated with the triglyceride level and inversely correlated with the HDL cholesterol level; however, there was no correlation with the total cholesterol or LDL cholesterol levels. Another study based on the Chinese Han population also reported similar results [33]. In Fukushima et al.’s study, the LDL cholesterol level showed weak positive correlation with the plasma BCAA concentration in the Japanese population [34]. These findings suggest that BCAA is a possible predictor of dyslipidemia.

Considering that SOB is an indicator of gingivitis or the early phase of periodontitis [21,27], and that BCAA is a predictor of insulin resistance in the early phase of diabetes [7,15], our findings suggest that the early phases of periodontitis and type 2 diabetes are correlated. While further studies are required to obtain evidence regarding the usefulness of periodontal therapy for improving the glycemic metabolism in patients with type 2 diabetes [35], our findings suggest that controlling gingival conditions is important to maintain sound glycemic metabolism by maintaining plasma BCAA concentrations.

This study had some limitations. First, the sample size was small, as there were only 64 participants; a larger sample size is needed to adjust for other confounders. Second, all participants were Japanese. Higher BCAA levels before and after the development of type 2 diabetes were observed in Caucasian and Asian ethnic groups, but not in African Americans [36]. Future studies should include other ethnic groups to generalize our findings. Third, in addition to oral microbes, dietary sources may also affect plasma BCAA levels. None of the participants had taken BCAA supplements. To assess the effect of dietary BCAA on the plasma BCAA concentration, a questionnaire nutrition survey, such as a food-frequency questionnaire [37], might be required. Fourth, as this was a cross-sectional study, a causal association between SOB and plasma BCAA levels could not be elucidated. Further cohort studies on the relationship between SOB and plasma BCAA levels are required.

## 5. Conclusions

We found a positive correlation between SOB and plasma BCAA levels in middle-aged Japanese individuals. SOB was a possible predictor for elevated plasma BCAA levels after adjusting for age, sex, and BMI.

## Figures and Tables

**Table 1 ijerph-19-08930-t001:** Characteristics of participants according to the SOB-test score.

	SOB		*p*-Value *
	Negative (n = 43)	Positive (n = 21)	
Number of teeth (*m*)	27 (16–28)	26 (16–28)	0.086
BOP (*m*)	0 (0–13)	2 (0–11)	0.010
%BOP (*m*)	0 (0–46.4)	7.4 (0–41.2)	0.007
≥4 mm PPD (*m*)	0.5 (0–15)	2 (0–20)	0.001
Age (*m*)	51 (2–74)	56 (5–78)	0.296
Sex [*n* (%)]			0.292
Male	12 (27.9)	8 (38.1)	
Female	31 (72.1)	13 (61.9)	
BMI (*m*; kg/m^2^;)	22.3 (17.8–32.5)	22.4 (19.1–31.5)	0.415
Total cholesterol (*m*; mg/dL)	243 (163–311)	230 (157–292)	0.482
HDL cholesterol (*m*; mg/dL)	74 (40–133)	68 (38–119)	0.189
LDL cholesterol (*m*; mg/dL)	146 (46–211)	143 (90–206)	0.975
Triglyceride (*m*; mg/dL)	78 (38–442)	87 (48–262)	0.285
Glucose (*m*; mg/dL)	92 (75–121)	96 (81–113)	0.175
HbA1c (*m*; %)	5.3 (4.8–7.2)	5.5 (4.9–7)	0.269
BCAAs (*m*; μmol/L)	432 (307–665)	477 (400–658)	0.034

*: Mann–Whitney U test for continuous variables (m), chi square test for categorical variables (n). SOB—salivary occult blood; BOP—bleeding on probing; PPD—periodontal pocket depth; BMI—body mass index; HDL—high-density lipoprotein; LDL—low-density lipoprotein; BCAA—branched-chain amino acid.

**Table 2 ijerph-19-08930-t002:** Correlation coefficients between BCAAs and diabetes-related factors.

	Correlation Coefficient	*p*-Values *
Glucose	0.22	0.101
HbA1c	0.30	<0.001
BMI	0.21	0.112
Total cholesterol	0.04	0.772
HDL cholesterol	−0.35	0.006
LDL cholesterol	0.18	0.170
Triglyceride	0.44	0.001

*: Spearman rank-order correlation coefficient. BMI—body mass index; HDL—high-density lipoprotein; LDL—low-density lipoprotein.

**Table 3 ijerph-19-08930-t003:** Linear regression analysis between SOB and BCAAs.

Crude Model			Adjusted Model *		
B ± SE	β	*p*-Values	B ± SE	β	*p*-Values
36.9 ± 12.6	0.24	0.004	26.7 ± 11.4	0.17	0.020

B—partial regression coefficient; SE—standard error; β—standardizing coefficient. *—Adjusted for sex, age, and BMI.

## Data Availability

The data are not publicly available, because participants of the study participated under the condition that the data were not available by other researchers.

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
