# Peer review of "Correlation of Salivary Occult Blood with the Plasma Concentration of Branched-Chain Amino Acids: A Cross-Sectional Study"

_ijerph, 2022, doi:10.3390/ijerph19158930_

Round 1

Reviewer 1 Report

Dear Authors

I appreciate your efforts in this study.

The present manuscript , it was evaluated whether could  indicators of periodontitis  ( SOB ) and  plasma BCAA in  middle-aged individuals have relation  together in the middle aged  individuals?

 I generally liked this topic. The subject is interesting and may lead to clinical implications in the future, also, I think the present manuscript has novelty. 

Only , there is some of minor shortcomings,

1.     Please write in detail about the method of determining the sample size

2.     I do not agree with the final sentence of the conclusion, I think this subject is  not the result of this study. And so I suggest revising the conclusion.    

      With regards

Author Response

Reply to comments

Reviewer 1

The present manuscript, it was evaluated whether could indicators of periodontitis (SOB) and plasma BCAA in middle-aged individuals have relation together in the middle-aged individuals?

 I generally liked this topic. The subject is interesting and may lead to clinical implications in the future, also, I think the present manuscript has novelty.

Only, there is some of minor shortcomings,

  1. Please write in detail about the method of determining the sample size

Reply: Thank you for your comment. The sample size was not estimated at the start of the study. The following subsection was added to the Materials and Methods section:

2.4. Sample size

The sample size was not estimated at the start of the study.

  1. I do not agree with the final sentence of the conclusion, I think this subject is not the result of this study. And so I suggest revising the conclusion.

Reply: As the reviewer’s suggestion, the last sentence of the conclusion was deleted.

Reviewer 2 Report

Dear authores some considerations should be modified, but in general congratulations for your effort .

Author Response

Reviewer 2

I congratulate on your work, but I would like you to make the following modifications I suggest below, thank you:

  1. The tile does not fit with the work, actually Salivary Occult Blood is the test what will be used to detect the concentration of Plasma Branched-Chain Amino Acid, But in fact the final goal is to see this test as a possible predictor of Diabetes type II. I suggest removing SOB from the title and adapting it.

Reply: Thank you for the comment. The SOB test was used to classify the periodontal condition. Plasma BCAA concentration was determined using venous blood drawn at the cubital fossa, not SOB. The aim of the study was to evaluate the correlation between SOB and the BCAA concentration. Thus, the word “SOB” is required in the title.

  1. Introduction: you should be included some information about SOB, since it is the focus of the research and nothing has been explained about the test, even in material and methods, what is measures and how it is work.

Reply: Thank you for the suggestion. The background of the SOB test was added in the Introduction section (lines 58–61). The test procedure has been described in the Materials and Methods section (lines 89–97)

  1. Material and method:
  2. Inclusion criteria are missing, they are very important and are not mentioned, it is even commented on the text that there are no smoking patients and that should be one of them

Reply: The inclusion criterion of the study was age ≥40 years. The sentence was added in lines 69. Smoking habit was added in data collection section (line 88).

  1. In Data Collection section, it is necessary to indicate which index are used (names) and references.

Reply: Thank you for the comment. However, there is no index to report.

  1. In Statistical Analysis section, delete the last sentence, STROBE, and add it at the end of the first section “study setting and study population”, and add the STROBE questionnaire as anex attachment.

Reply: According to the reviewer’s suggestion, the statement regarding STROBE was moved from the statistical analysis section to the study setting and study population. The STROBE checklist has been attached as a supplemental file.

  1. Results: all ok

Reply: Thank you for your comment.

  1. Discussion: Add references to the first three sentences of the paragraph (185-187lines) “Considering that SOB is an indicator of gingivitis ……..

Reply: According to the reviewer’s suggestion, the references for the relationships between SOB and periodontitis and between BCAA and insulin resistance were added.

  1. Conclusions: the last sentence should be deleted as it does not take into account the hypothesis of profesional oral higiene and care.

Reply: According to the reviewer’s suggestion, we have deleted the last sentence of the conclusion.

Round 2

Reviewer 2 Report

Please just the last modifications, It is necesary to coment the INDEXES ( peridontal) you have uses and the references at M&M section.

Author Response

Please just the last modifications, It is necesary to coment the INDEXES ( peridontal) you have uses and the references at M&M section.

Reply: According to the reviewer’s suggestion, the sentence “To distinguish periodontal disease, the number and ratio (%) of teeth with BOP and the number of teeth with PPD ≥ 4 mm were assessed [21].” was added.
